Journal of
open psychology data

# The Values in Crisis Project: A Three-Wave Panel Study in Germany and the United Kingdom

DATA PAPER

CHRISTIAN WELZEL (iD)

KLAUS BOEHNKE (iD)

JAN DELHEY (iD)

FRANZISKA DEUTSCH (iD)

JAN EICHHORN (iD)

ULRICH KÜHNEN (iD)

GEORGI DRAGOLOV

STEPHANIE HESS (iD)

MANDI LARSEN (iD)

*Author affiliations can be found in the back matter of this article

]u[ ubiquity press

## ABSTRACT

This article introduces the data from the Values in Crisis project conducted in Germany and the United Kingdom. The project seized the COVID-19 pandemic as a natural experiment to investigate whether, how and to what extent people's moral values change as a result of a disruptive event of massive order and global scale. An online panel survey measured individuals' experiences with COVID-19, moral values, personality traits and social orientations at three different stages throughout the pandemic: at its onset (Wave 1: April–May 2020), one year later amidst the pandemic (Wave 2: February–March 2021), and two years later towards its end (Wave 3: February–April 2022). The samples for Wave 1 were drawn using quota sampling along gender, age group, level of education, and country region for the population aged 16 and above in Germany ($N_{DE,W1}$ = 2,005), and 18 and above in the UK ($N_{UK,W1}$ = 2,033). The samples for Wave 2 consist of re-contacted participants at a retention rate of 63.99% for Germany ($N_{DE,W1–2}$ = 1,283) and 56.57% for the UK ($N_{UK,W1–2}$ = 1,150). The samples for Wave 3 comprise of re-contacted participants at a retention rate of 43.74% in Germany ($N_{DE,W1–3}$ = 877) and 37.73% in the UK ($N_{UK,W1–3}$ = 767) as well as newly recruited participants ($N_{DE,W3}$ = 381, $N_{UK,W3}$ = 461). The data can be used for various secondary analyses on the topics covered in the survey.

**CORRESPONDING AUTHOR:**

**Klaus Boehnke**

Constructor University, Bremen, Germany

kboehnke@constructor. university

**KEYWORDS:**
COVID-19 pandemic; value change; panel study; United Kingdom; Germany

**TO CITE THIS ARTICLE:**
Welzel, C., Boehnke, K., Delhey, J., Deutsch, F., Eichhorn, J., Kühnen, U., Dragolov, G., Hess, S., & Larsen, M. (2024). The Values in Crisis Project: A Three-Wave Panel Study in Germany and the United Kingdom. *Journal of Open Psychology Data,* 12: 1, pp. 1–11. DOI: https://doi.org/10.5334/jopd.89

# (1) BACKGROUND

Moral values are of critical relevance for societal well-being as they determine how people behave in social settings. The aggregate distribution of values in a society shapes prevalent patterns of human behaviour, which guide the overall development of that society (Caprara et al., 2006; Schwartz et al., 2014; Vecchione et al., 2015).

There is consensus in the literature that people's moral values are shaped during the so-called formative phase of socialization (Arnett, 2015), which is likely to be completed by about the age of twenty-five. Moral values internalized by then are considered to be stable for the remainder of an individual's lifetime (Milfont et al., 2016). Even if momentary adjustments do occur in response to situational changes, these situation-based adjustments usually oscillate around stable personal baselines. For this reason, ground-breaking value change on a societal level only proceeds at a glacial pace, either through generational replacement or through synchronic—albeit slight—up- or downward slopes in individual value baselines. Thus, rapid value change, cannot happen under 'normal' circumstances. By contrast, it is an open question whether and to what extent the usual stability of value orientations and the gradualness of their change continue during exceptionally disruptive times, that is, when a crisis of massive proportions suddenly hits all members of society. More specifically, can an incisive crisis—such as the COVID-19 pandemic—cause a lasting tectonic shift in the intercepts and slopes of people's personal value baselines, with baseline levels simultaneously leaping up- or downward and slope angles being twisted steeper in direction? The current literature does not offer conclusive empirical answers to this question.

The Values in Crisis project seized the COVID-19 pandemic as a truly unique opportunity—a kind of natural experiment. The data described in this article were collected in order to study empirically how people's moral values behave during times of crises. The COVID-19 pandemic is, beyond any doubt, the most dramatic social crisis since World War II. It started suddenly and unexpectedly to progress rapidly on a global scale, thereby threatening the lives and existence of virtually each and every individual. The general sense of disruption was strengthened by the unprecedented response of governments that involved major incisions into people's everyday lives: essential life domains such as mobility, employment, education and access to healthcare were affected by unparalleled restrictions (Hale et al., 2021).

The major disruption caused by the COVID-19 pandemic makes it possible to address a number of unresolved research questions: Do people change their values under the imprint of this crisis? If so, how massive are these changes? Which direction do these changes take? Do people's moral values revert back to their old setpoint or does the crisis rather leave a lasting impact? In case of the latter, do only the intercept levels of people's value trajectories experience a tectonic shift; or are also the slope directions of these trajectories twisted steeper in angle?

Informed by various versions of existential insecurity theories (Inglehart, 1997; Inglehart & Welzel 2005; Murray & Schaller, 2012; Pyszczynski et al., 2003), an intuitive hypothesis suggests that a crisis-induced sudden rise in existential anxieties would cause a shift from emancipative to protective values. A protective value shift would drive people to emphasize security, order, authority, uniformity and conformism (see for example Daniel et al., 2022). Consequently, preferences for emancipative values such as out-group trust, tolerance for diversity and in-group transcending solidarity would weaken, making the appeal of authoritarian government stronger. Moreover, the hypothesized value shift in the protective direction may be more pronounced for individuals who have been struggling under the crisis-induced reduction in their capabilities and resources to act freely in comparison to those who have been nevertheless thriving. If enduring, the consequence of such mentality shifts for public support for democracy would be dire. Figure 1 provides a stylized depiction of this scenario.

Alternatively, the universal character of the COVID-19 pandemic in threatening everyone, irrespective of social class, ethnicity or religion, may strengthen a generalized sense of humanity in people (Beck, 1992; Buchan et al., 2011). In this case, out-group trust, tolerance for diversity and transcendent solidarity might increase and, hence, diminish the appeal of authoritarian government. In this case, public support for democracy would not suffer. In fact, the experience of daily restrictions in civil liberties might even strengthen the overall appreciation of democratic freedoms.

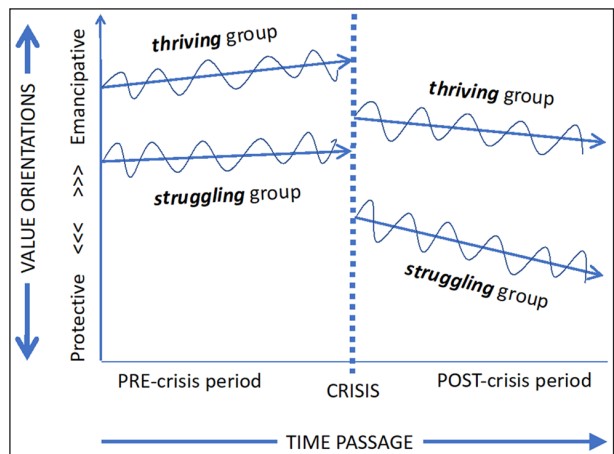

**Figure 1** Crisis-induced drop in intercept levels and downturn twist in slope angles of value baselines among thriving and struggling population segments.

As a third possibility, both effects might exist but affect different groups of people defined along certain characteristics, most notably personality traits. To give an example, people who score high on neuroticism in the Big Five personality framework (Rammstedt & John, 2007) might experience a particularly powerful value shift into a protective direction and diminish their out-group trust, tolerance for diversity and transcendent solidarity. By contrast, people who score high on openness in the Big Five personality framework might experience a value shift into the exact opposite direction under the COVID-19 pandemic and increase their out-group trust, tolerance for diversity and transcendent solidarity. This type of mentality shifts of different groups in opposite directions would, in turn, result in cultural polarization and greater potential for disruptive ideological conflict. The prospects for the healthiness of democracy would again be rather bleak in this scenario.

The data collected for the Values in Crisis project can help uncover which of the mechanisms outlined above have been at play. We introduce the data in the next sections of this article.

# (2) METHODS

In order to address the research questions, the Values in Crisis project made exclusive use of quantitative methodology. More specifically, the project involved

the collection of primary cross-culturally comparative longitudinal survey data. Table 1 provides critical details on the study design.

## 2.1 STUDY DESIGN

The Values in Crisis project outlined here entailed a comparison of Germany and the United Kingdom as an ideal mixture of fundamental similarities (thereby allowing for comparability) and important differences (thereby allowing for variation). Both countries are highly developed post-industrial societies, representing the two largest populations and economies in Europe (IMF, 2018) as well as mature and stable democracies. On the other hand, there are notable differences in the healthcare systems of both countries (Reibling et al., 2019) and the initial response of the authorities to the pandemic (Ziegler, 2020).

The explicit research focus on a potential change in people's moral values required studying the same people throughout different stages of the pandemic. Three such stages were considered relevant: First, at the onset of the pandemic (Wave 1); second, amidst the pandemic (Wave 2); and third, towards its end (Wave 3). In both the United Kingdom and Germany, each wave of data collection was carried out by *bilendi GmbH*, an opinion research company specialized in online data collection.

Following a longstanding tradition in value research (Inglehart, 1977, 1997; Schwartz 1992), the project measured moral values via individual self-reports in a survey framework. Given the contact restrictions

| COUNTRY | WAVE 1 | | WAVE 2 | | WAVE 3 | |
|---|---|---|---|---|---|---|
| | **GERMANY** | **UK** | **GERMANY** | **UK** | **GERMANY** | **UK** |
| Language | German | English | German | English | German | English |
| Mode | standardized questionnaire, CASI | | standardized questionnaire, CASI | | standardized questionnaire, CASI | |
| Duration (minutes) | | | | | | |
| Mean | 19.10 | 18.98 | 18.60 | 19.71 | 21.79 | 24.17 |
| Standard deviation | 40.11 | 61.23 | 30.41 | 54.70 | 36.15 | 74.67 |
| Field phase | | | | | | |
| Start | 24/04/2020 | 29/04/2020 | 15/02/2021 | 23/02/2021 | 16/02/2022 | 16/02/2022 |
| End | 10/05/2020 | 19/05/2020 | 01/03/2021 | 15/03/2021 | 28/04/2022 | 28/04/2022 |
| Target population | adult country residents | | adult country residents | | adult country residents | |
| Recruitment | online panel, invitation by email | | online panel, invitation by email | | online panel, invitation by email | |
| Incentive | duration based, cash or tokens | | duration based, cash or tokens | | duration based, cash or tokens | |
| Sampling strategy | quota sample | | recontact | | recontact, quota for the new | |
| Sample size (total) | 2,005 | 2,033 | 1,283 | 1,150 | 1,258 | 1,228 |
| Panelists | | | 1,283 | 1,150 | 877 | 767 |
| Newly recruited | | | 0 | 0 | 381 | 461 |
| Retention rate | | | 63.99% | 56.57% | 43.74% | 37.73% |

**Table 1** Methodological fact sheet on the sample.

Welzel et al. *Journal of Open Psychology Data* DOI: 10.5334/jopd.89 

throughout the three waves of data collection, an online survey (CASI: Computer-Assisted Self-Interview) was the only viable option. In each wave, the survey employed a fully standardized questionnaire consisting of closed-ended items only. Participants took, on average, about twenty minutes to complete questionnaire (see Table 1 for further details).

As a side note, the Values in Crisis – Germany/UK project inspired extensive international collaboration. This resulted in the so-called Values in Crisis International project (Aschauer et al., 2021) which fielded the first wave in altogether 18 countries worldwide: Austria, Brazil, Chile, China, Colombia, Georgia, Germany, Greece, Hong Kong, Italy, Japan, Kazakhstan, South Korea, Maldives, Poland, Russia, Sweden, United Kingdom.[1] The international data are beyond the scope of the present article.

## 2.2 TIME OF DATA COLLECTION

The first wave of the survey was fielded from April 24 to May 19, 2020; the second wave from February 15 to March 15, 2021; and the third wave from February 16 to April 28, 2022.

## 2.3 LOCATION OF DATA COLLECTION

Each wave of the survey was fielded in Germany and the United Kingdom. The chosen sampling strategy (see 2.4) allows for a regionalization of the collected data in each wave. As to Germany, the respective samples can be broken down to the sixteen federal states (*Bundesländer*) of the Federal Republic. A regional comparison with respect to the borders of former West and East Germany is also possible, with an important qualification: The data collected for Berlin do not allow for a differentiation between the former West and East districts of Germany's capital city Berlin. As to the UK, the respective samples can be broken down to the four constituent countries of the UK (England, Wales, Scotland, Northern Ireland) and the nine English regions. More details can be seen in Table 2a for Germany and Table 2b for the United Kingdom.

## 2.4 SAMPLING, SAMPLE AND DATA COLLECTION

Given the project aims, the ideal sample would, undoubtedly, be a representative sample drawn from a target population of all adults residing in the respective country. The online mode of data collection, however, precluded a random-sampling technique (e.g. random probability or stratified random sampling). Respondents were instead drawn from the pools of potential participants (online panels) available to the data collecting company. Participation in each wave was incentivized based on the

| REGION | TARGET QUOTA | WAVE 1: CROSS-SECTION | | WAVE 2: CROSS-SECTION PANEL: WAVES 1 AND 2 | | WAVE 3: CROSS-SECTION | | PANEL: WAVES 1, 2 AND 3 | |
|---|---|---|---|---|---|---|---|---|---|
| | % | N | % | N | % | N | % | N | % |
| Schleswig-Holstein | 3.37 | 68 | 3.39 | 45 | 3.51 | 43 | 3.42 | 34 | 3.88 |
| Hamburg | 2.18 | 43 | 2.14 | 28 | 2.18 | 28 | 2.23 | 19 | 2.17 |
| Niedersachsen | 9.50 | 191 | 9.53 | 114 | 8.89 | 123 | 9.78 | 80 | 9.12 |
| Bremen | 0.79 | 16 | 0.80 | 12 | 0.94 | 13 | 1.03 | 7 | 0.80 |
| Nordrhein-Westfalen | 21.58 | 437 | 21.80 | 268 | 20.89 | 258 | 20.51 | 177 | 20.18 |
| Hessen | 7.33 | 149 | 7.43 | 94 | 7.33 | 92 | 7.31 | 69 | 7.87 |
| Rheinland-Pfalz | 5.05 | 98 | 4.89 | 62 | 4.83 | 66 | 5.25 | 41 | 4.68 |
| Baden-Württemberg | 13.07 | 260 | 12.97 | 173 | 13.48 | 156 | 12.40 | 107 | 12.20 |
| Bayern | 15.25 | 306 | 15.26 | 200 | 15.59 | 195 | 15.50 | 147 | 16.76 |
| Saarland | 1.29 | 26 | 1.30 | 14 | 1.09 | 17 | 1.35 | 9 | 1.03 |
| Berlin | 4.36 | 86 | 4.29 | 57 | 4.44 | 51 | 4.05 | 38 | 4.33 |
| Brandenburg | 3.07 | 62 | 3.09 | 48 | 3.74 | 41 | 3.26 | 30 | 3.42 |
| Mecklenburg-Vorpommern | 2.08 | 42 | 2.09 | 34 | 2.65 | 28 | 2.23 | 21 | 2.39 |
| Sachsen | 5.25 | 106 | 5.29 | 59 | 4.60 | 71 | 5.64 | 44 | 5.02 |
| Sachsen-Anhalt | 3.07 | 59 | 2.94 | 37 | 2.88 | 38 | 3.02 | 28 | 3.19 |
| Thüringen | 2.77 | 56 | 2.79 | 38 | 2.96 | 38 | 3.02 | 26 | 2.96 |
| Total | 100.00 | 2,005 | 100.00 | 1,283 | 100.00 | 1,258 | 100.00 | 877 | 100.00 |

**Table 2a** Relative and absolute frequencies (unweighted) of cases in samples from Germany by region and wave in cross-section and panel.

| REGION | TARGET QUOTA | WAVE 1: CROSS-SECTION | | WAVE 2: CROSS-SECTION PANEL: WAVES 1 AND 2 | | WAVE 3: CROSS-SECTION | | PANEL: WAVES 1, 2 AND 3 | |
|---|---|---|---|---|---|---|---|---|---|
| | % | N | % | N | % | N | % | N | % |
| *England* | | | | | | | | | |
| *North East* | 4.15 | 79 | 3.89 | 51 | 4.43 | 53 | 4.32 | 35 | 4.56 |
| *North West* | 11.15 | 234 | 11.51 | 138 | 12.00 | 133 | 10.83 | 85 | 11.08 |
| *Yorkshire and the Humber* | 8.35 | 169 | 8.31 | 103 | 8.96 | 102 | 8.31 | 71 | 9.26 |
| *East Midlands* | 7.20 | 147 | 7.23 | 90 | 7.83 | 92 | 7.49 | 62 | 8.08 |
| *West Midlands* | 8.75 | 176 | 8.66 | 82 | 7.13 | 92 | 7.49 | 49 | 6.39 |
| *East of England* | 9.25 | 184 | 9.05 | 109 | 9.48 | 107 | 8.71 | 72 | 9.39 |
| *London* | 12.80 | 264 | 12.99 | 136 | 11.83 | 164 | 13.35 | 90 | 11.73 |
| *South East* | 13.60 | 281 | 13.82 | 162 | 14.09 | 172 | 14.01 | 102 | 13.30 |
| *South West* | 8.50 | 174 | 8.56 | 86 | 7.48 | 112 | 9.12 | 63 | 8.21 |
| *Wales* | 4.90 | 99 | 4.87 | 54 | 4.70 | 60 | 4.89 | 38 | 4.95 |
| *Scotland* | 8.55 | 175 | 8.61 | 107 | 9.30 | 113 | 9.20 | 75 | 9.78 |
| *Northern Ireland* | 2.80 | 51 | 2.51 | 32 | 2.78 | 28 | 2.28 | 25 | 3.26 |
| *Total* | 100.00 | 2,033 | 100.00 | 1,150 | 100.00 | 1,228 | 100.00 | 767 | 100.00 |

**Table 2b** Relative and absolute frequencies (unweighted) of cases in samples from the United Kingdom by region and wave in cross-section and panel.

incentive scheme of the data collecting company. The incentive scheme considers the duration of the survey and offers a choice between cash or loyalty tokens. Because these individuals have volunteered to register as potential participants in online surveys and are incentivized, the samples drawn unavoidably entail a certain degree of bias with respect to online accessibility, computer literacy, interest for material rewards, or affinity for and experience with surveys. These aspects can potentially reflect specific socio-economic backgrounds. In order to reduce biases along such lines, quota sampling was employed in the first wave of data collection. The quotas were defined along gender, age group, level of education and region for the respective country populations (of age 16 and above in Germany, and 18 and above in the United Kingdom) based on the information provided by the national statistical offices. For more details on the target quotas and realized samples in each wave, see Tables 2a and 2b (on regions) and Table 3 (on further relevant socio-demographics). Participants were recruited by invitation via the interface of the data collecting company. The realized samples in the first wave consist of 2,005 respondents from Germany and 2,033 respondents from the United Kingdom.

The sample for Wave 2 consists exclusively of participants from Wave 1: Respondents who participated in the first wave were re-contacted and invited to participate in the second wave without quota-based screening. The achieved samples in Wave 2 consist of 1,283 respondents from Germany and 1,150 respondents from the United Kingdom. The achieved retention rates amount to 63.99% for Germany and 56.57% for the

United Kingdom (see Tables 1, 2a, 2b and 3 for further details).

The samples for Wave 3 consist of re-contacted and newly recruited participants (refresher sample). The latter were invited only after the pool of to-be-re-contacted participants had been exhausted and were subjected to quota-based screening. This was done in order to allow for more detailed segmentations in cross-sectional analyses of Wave 3 for which the size of the longitudinal sample could have proven limited. The resulting panel data for Waves 1 to 3 include 877 respondents from Germany and 767 respondents from the United Kingdom (which was higher than originally expected). This makes a retention rate of 43.74% in Germany and 37.73% in the United Kingdom, comparing Waves 1 and 3. Joint with the refresher sample, the cross-sectional data for Wave 3 consist of 1,258 respondents from Germany and 1,228 respondents from the United Kingdom.

The discrepancies between the target quotas and the achieved relative frequencies along the categories of the relevant socio-demographic characteristics (see Tables 2a, 2b and 3) can be corrected for by applying data weights. The weights were calculated by the research team using the raking procedure (iterative proportional fitting; Kolenikov, 2014). Data weights are available for cross-sectional analyses of each wave as well as for longitudinal analyses.

## 2.5 MATERIALS/SURVEY INSTRUMENTS
The thematic foci of the questionnaire encompass the physical and psychological experience of COVID-19,

| CHARACTE-RISTIC | TARGET QUOTA | | WAVE 1: CROSS-SECTION | | WAVE 2: CROSS-SECTION PANEL: WAVES 1 AND 2 | | WAVE 3: CROSS-SECTION | | PANEL: WAVES 1, 2 AND 3 | |
|---|---|---|---|---|---|---|---|---|---|---|
| | GERMANY | UK | GERMANY | UK | GERMANY | UK | GERMANY | UK | GERMANY | UK |
| *Sex* | | | | | | | | | | |
| *Male* | 50.1 | 48.3 | 50.12 | 48.25 | 50.74 | 54.43 | 49.05 | 49.02 | 53.25 | 55.54 |
| *Female* | 49.9 | 51.7 | 49.88 | 51.75 | 49.26 | 45.57 | 50.95 | 50.98 | 46.75 | 44.46 |
| *Diverse* | 0.0 | 0.0 | 0.00 | 0.00 | 0.00 | 0.00 | 0.00 | 0.00 | 0.00 | 0.00 |
| *Age group* | | | | | | | | | | |
| *16–24 years* | 13.2 | 11.9 | 13.32 | 12.00 | 5.92 | 5.30 | 8.43 | 8.06 | 3.99 | 4.30 |
| *25–34 years* | 15.9 | 17.0 | 16.01 | 17.71 | 12.70 | 14.17 | 14.07 | 16.21 | 10.38 | 12.91 |
| *35–44 years* | 17.9 | 17.6 | 17.91 | 16.58 | 17.61 | 15.57 | 17.01 | 17.92 | 17.79 | 14.86 |
| *45–54 years* | 21.7 | 17.6 | 21.75 | 17.51 | 25.02 | 19.83 | 22.10 | 17.92 | 27.02 | 18.90 |
| *55–64 years* | 16.8 | 14.9 | 16.76 | 15.05 | 20.11 | 18.00 | 19.71 | 15.64 | 21.44 | 19.04 |
| *65+ years* | 14.5 | 21.0 | 14.26 | 21.15 | 18.63 | 27.13 | 18.68 | 24.27 | 19.38 | 29.99 |
| *Education* | | | | | | | | | | |
| *Low* | 19.5 | 19.1 | 19.25 | 21.25 | 21.04 | 35.83 | 18.60 | 28.99 | 20.75 | 20.34 |
| *Middle* | 55.3 | 40.3 | 55.36 | 38.51 | 55.11 | 21.30 | 57.23 | 27.93 | 55.76 | 41.33 |
| *High* | 25.2 | 40.6 | 25.39 | 40.24 | 23.85 | 42.87 | 24.17 | 43.08 | 23.49 | 38.33 |

**Table 3** Demographic characteristics (relative frequencies) by country in cross-section and panel.

moral values, personality traits, social orientations and ideological leaning as well as subjective well-being. It further includes basic socio-demographic information on gender, age, education, income, residence, religion and ethnicity.

Most of the key instruments were directly adopted from established large-scale comparative surveys with a proven track record such as the European Social Survey (ESS, 2020), the World Values Survey (Inglehart et. al., 2018) and the European Quality of Life Survey.[2] This approach not only guarantees the validity of the measures, but also makes it possible to use the data from these surveys as benchmarks. Some further items were adapted from these surveys with only slightly modified wording. The questionnaire also included novel items developed specifically for the research purposes of the Values in Crisis project, e.g. those addressing the unprecedented COVID-19 situation. Due to the sudden onset of the COVID-19 pandemic and the speed at which related events were unfolding (e.g. government response), we were compelled to prefer an expedient start of the data collection phase to a full-fledged pilot test phase aimed at establishing the psychometric properties of the modified and novel items. The latter could have turned out time intensive and, thereby, rendered our research 'obsolete'. Studies employing our data should, therefore, consider the potential methodological problems arising from this compromise (see Flake and Fried, 2020).

Table 4 provides an overview of the wide array of topics covered in each wave of the survey and cross-references the questionnaire items. The careful reader may notice that some topics were added or discontinued in subsequent waves. The decision as to measure additional aspects addressed in an earlier wave or to measure aspects not previously addressed was guided by their research relevance in light of the course of events around the COVID-19 pandemic, or by considerations of space and cost. For example, personality traits were measured only in Wave 1 as they are generally considered to be stable characteristics of individuals. Items on attitudes towards vaccines or respondents' vaccination status were included in Waves 2 and 3, i.e. only when they became relevant.

For further details, please refer to the data documentation.

## 2.6 QUALITY CONTROL

Several steps were undertaken in order to ensure the quality of the data collected. These involved the identification of 'speeders' (respondents who complete the questionnaire way too quickly in comparison to the rest) and 'straightliners' (respondents who select one and the same answering option in a matrix of multiple items). Quality control involved also various plausibility checks with the aim to identify respondents who provide contradicting or highly implausible answers. To name a few examples: a respondent selects totally

| TOPIC | WAVE 1 | WAVE 2 | WAVE 3 |
|---|---|---|---|
| *Experience of COVID-19* | | | |
| *Personal health consequences* | Q10 | Q10 | Q10 |
| *Socio-economic consequences* | Q11 | Q11 | Q11 |
| *Worries about self and society* | Q12–13, Q17 | Q12–13, Q17 | Q12–13 |
| *Evaluation of government measures* | Q14 | Q14 | Q14, Q65 |
| *Evaluation of own and others' behavior* | Q15 | Q15, Q45, Q47 | Q15, Q45, Q47 |
| *Solidarity* | Q16 | Q16 | Q16 |
| *Conspiracy theory receptivity* | Q19 | Q19, Q48 | Q19, Q48, Q61 |
| *Importance of freedom vs. health protection* | | Q46 | Q46, Q56 |
| *Attitudes towards vaccines, vaccination status* | | Q49–51 | Q50–51, Q57–59 |
| *Moral values* | | | |
| *Schwartz values* | Q21 | Q21 | Q21 |
| *Sacred-secular, patriarchal-emancipative values* | Q22–29 | Q22–29 | Q22–29 |
| *Personality traits* | | | |
| *BIG-Five* | Q30 | | |
| *Davis' empathy scale* | Q31 | | |
| *Social orientations and ideological leaning* | | | |
| *Institutional trust* | Q20, Q33 | Q20, Q33 | Q20, Q33 |
| *Interpersonal trust* | Q35–36 | Q35 | Q35 |
| *Attitudes towards migration and diversity* | Q37–38 | Q37–38 | Q37–38 |
| *Country aims and political priorities* | Q32, Q39 | Q32, Q39 | Q32, Q39 |
| *Political orientation* | Q40 | Q40 | Q40 |
| *Populism* | | Q52 | Q52 |
| *Perception of social cohesion and tensions* | | | Q62, Q64 |
| *Subjective well-being* | | | |
| *Depression and anxiety (PHQ-4), loneliness* | Q18 | Q18 | Q18 |
| *Overall life and domain-level satisfaction* | Q34 | Q34 | Q34 |
| *Social exclusion* | | | Q60 |
| *Socio-demographic section* | | | |
| *Gender* | Q1 | Q1 | Q1 |
| *Age* | Q2 | Q2 | Q2 |
| *Family situation (marital status, children)* | Q3–4 | Q3–4 | Q3–4 |
| *Level of education* | Q5 | Q5 | Q5 |
| *Income* | Q6 | Q6 | Q6 |
| *Household composition* | Q7 | Q41–43 | Q41–43 |
| *Size of place of residence* | Q8 | | Q8 |
| *Region of residence* | Q9 | Q9 | Q9 |
| *Employment situation* | | Q44 | Q44 |

**Table 4** Survey contents by wave with cross-references to questionnaire items.

opposing answering options for statements with similar content; a respondent selects unlikely combinations of categories in the socio-demographic section such as young age and retired status. In addition, the data were inspected for the presence of doublets (multiple occurrences of cases with identical identifiers in a cross-section) and implausible changes in the socio-demographics in the longitudinal scenario. The latter may occur when different members of the household happen to participate in the different waves of the study.

Further details on the steps for quality control can be obtained from the opinion research company, a corporate member of the European Society for Opinion and Marketing Research (ESOMAR).

### 2.7 DATA ANONYMISATION AND ETHICAL ISSUES

Participants were informed about the general research aims of the project and asked to provide their informed consent prior to responding to the items. Data were anonymised by the opinion research company and delivered to the research team in anonymous form only.

Ethical issues were carefully discussed among the Principal Investigators and with the opinion research company. As per the guidelines of the German Research Foundation (DFG) on research projects in the Humanities and Social Sciences, a statement by an ethics committee was not required.

### 2.8 EXISTING USE OF DATA

The following peer-reviewed journal articles have been published so far:

Delhey, J., Hess, S., Boehnke, K., Deutsch, F., Eichhorn, J., Kühnen, U., & Welzel, C. (2023). Life Satisfaction During the COVID-19 Pandemic: The Role of Human, Economic, Social, and Psychological Capital. *Journal of Happiness Studies*. https://doi.org/10.1007/s10902-023-00676-w

Delhey, J., Steckermeier, L., Boehnke, K., Deutsch, F., Eichhorn, J., Kühnen, U., & Welzel, C. (2023). Existential insecurity and trust during the COVID-19 pandemic: The case of Germany. *Journal of Trust Research*. https://doi.org/10.1080/21515581.2023.2223184

Eichhorn, J., Spöri, T., Delhey, J., Deutsch, F. & Dragolov, G. (2022). Reality bites: An Analysis of corona deniers in Germany over time. *Frontiers in Sociology, 7*: 974–972. https://doi:10.3389/fsoc.2022.974972

## (3) DATASET DESCRIPTION AND ACCESS

### 3.1 REPOSITORY LOCATION

The data and the accompanying documentation (questionnaires, codebook and methodological report) have been stored under the permanent identifier ZA7989 at the repository of GESIS – Leibniz Institute for the Social Sciences (Germany). Potential users can access the data and the documentation at: https://doi.org/10.4232/1.14148.

### 3.2 OBJECT/FILE NAME

All published datasets are stored in a zip-archive for the respective data format (see Section 3.4). Each archive contains the following files:

- ZA7989_w1_v1-0-0: Cross-sectional data from Wave 1
- ZA7989_w1–2_v1-0-0: Longitudinal data from Waves 1 and 2
- ZA7989_w1–3_v1-0-0: Longitudinal data from Waves 1, 2 and 3
- ZA7989_w3_v1-0-0: Cross-sectional data from Wave 3

The survey questionnaires are available in British English and German. The language version is indicated in the file name by the suffix "de" for German or "gb" for British English. All questionnaires are stored in a zip-archive containing the following files:

- ZA7989_q_[de/gb]_w1: Questionnaire for Wave 1
- ZA7989_q_[de/gb]_w2: Questionnaire for Wave 2
- ZA7989_q_[de/gb]_w3: Questionnaire for Wave 3

The codebooks accompanying the data are available in English only. These are stored in a zip-archive containing the following files:

- ZA7989_cod_w1: Codebook for cross-sectional data from Wave 1
- ZA7989_cod_w1–2: Codebook for longitudinal data from Waves 1 and 2
- ZA7989_cod_w1–3: Codebook for longitudinal data from Waves 1, 2 and 3
- ZA7989_cod_w3: Codebook for cross-sectional data from Wave 3

The category "Other documents" in the "Downloads" section offers access to the methodological report:

- ZA7989_mr.pdf

### 3.3 DATA TYPE

The datasets contain the raw primary data following data cleaning procedures and the respective weights calculated by the project team.

### 3.4 FORMAT NAMES AND VERSIONS

The datasets are available in SPSS, Stata (14 and above) and comma-separated format. The deposited archives containing the datasets have been indexed with the file extension for a dataset in the respective format: .sav (SPSS), .dta (Stata), .csv (comma-separated file).

The documentation (questionnaires, codebooks and methodological report) is available in pdf format.

### 3.5 LANGUAGE

Questionnaires are available in British English and German. Codebooks and datasets are available in British English only.

### 3.6 LICENSE AND LIMITS TO SHARING

The data have been released for academic research and teaching purposes only (category A). Access is granted free of charge to registered GESIS users. Registration with GESIS is also free of charge (https://login.gesis.org/). More information about GESIS' terms of use can be found on the following website: https://www.gesis.org/fileadmin/upload/Datenservices/Nutzungsbedingungen/2023-06-30_Usage_regulations.pdf.

### 3.7 PUBLICATION DATE

The data and the accompanying documentation were published in the repository on 21/09/2023.

### 3.8 FAIR DATA/CODEBOOK

Our data and their documentation as well as the data service provided by GESIS conform to the FAIR principles (Findable, Accessible, Interoperable, Reusable).

## (4) REUSE POTENTIAL

The data can be reused for many analytical purposes. In all the main topic areas targeted by the Values in Crisis survey — value orientations, socio-political attitudes and subjective well-being —, the potential of the data for both cross-sectional and longitudinal analyses is far from exhausted. Very likely, the main interest of re-users is how these constructs have evolved with changes in people's perception of the coronavirus pandemic. A major strength of the Values in Crisis survey in this context is that the main topics were surveyed using several proven constructs, such as value orientations not only according to the Schwartz value circle, but also according to Welzel and Inglehart's value concept. Second, there is considerable re-use potential in the inequality perspective: Which sub-populations have fared better and which have fared worse during the pandemic? In the academic and public debate, there are various portrayals of the pandemic in this regard, sometimes as reinforcing inequality,

sometimes as "equalizing" and "inequality-blind". A third potential perspective for re-using the data is that of social cohesion (cf. Larsen et al., 2023): Have people's attitudes and value orientations moved in a direction of solidarity or lack thereof? What is the state of consensus on attitudes and value orientations that are central to cohesion across the various stages of the pandemic?

By varying the units to be compared, one can also get a lot more out of the Values in Crisis data. Even though the study was originally intended as a two-country comparison only, the surveys can also be analysed on a sub-national level: the sixteen federal states of Germany, the four constituent countries of the UK or the nine regions of England. It would be especially profitable to combine the German data with pandemic-related characteristics of the federal states. A rich source of information in this context is the ZPID Lockdown Measures Dataset for Germany (Steinmetz et al., 2022). This dataset has tracked the implementation of anti-corona measures in the various federal states on a daily basis. Additional policy information of this kind can be used, for example, to examine whether the severity of measures is associated with the attitudes, values and well-being of the population. Moreover, the questionnaire of the first wave of the Values in Crisis project was fielded in further 16 countries worldwide (as mentioned above), which enables large-scale cross-culturally comparative studies — an opportunity that only few "Covid surveys" can provide.

Other information required for submission, not for review.

### NOTES

1  https://data.aussda.at/dataset.html?persistentId=doi:10.11587/LIHK1L.

2  https://www.eurofound.europa.eu/surveys/european-quality-of-life-surveys.

### ACKNOWLEDGEMENTS

We would like to thank the anonymous reviewers and the journal editors for their constructive feedback on earlier versions of the manuscript as well as the Data Acquisitions and Access officers at GESIS for their invaluable assistance with the data archiving process.

### FUNDING INFORMATION

The project "Values in Crisis: A Crisis of Values? Moral Values and Social Orientations under the Imprint of the Corona Pandemic" was funded by Volkswagen Foundation, Grant No. 99/127. Waves 2 and 3 of the VIC surveys in Germany and the United Kingdom were financed from this grant.

## COMPETING INTERESTS

The authors have no competing interests to declare.

## AUTHOR CONTRIBUTIONS

All authors contributed towards the writing of this manuscript and approved its final version.

## AUTHOR AFFILIATIONS

**Christian Welzel** orcid.org/0000-0002-1562-3580
Leuphana University, Lüneburg, Germany

**Klaus Boehnke** orcid.org/0000-0002-5435-4037
Constructor University, Bremen, Germany

**Jan Delhey** orcid.org/0000-0002-0475-8578
Otto von Guericke University Magdeburg, Germany

**Franziska Deutsch** orcid.org/0000-0001-9868-0988
Constructor University, Bremen, Germany

**Jan Eichhorn** orcid.org/0000-0003-4988-8891
University of Edinburgh, United Kingdom

**Ulrich Kühnen** orcid.org/0000-0001-9059-4719
Constructor University, Bremen, Germany

**Georgi Dragolov**
Constructor University, Bremen, Germany; Otto von Guericke University Magdeburg, Germany

**Stephanie Hess** orcid.org/0000-0003-1762-4461
Otto von Guericke University Magdeburg, Germany

**Mandi Larsen** orcid.org/0000-0001-5057-0085
Constructor University, Bremen, Germany

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

## PEER REVIEW COMMENTS

*Journal of Open Psychology Data* has blind peer review, which is unblinded upon article acceptance. The editorial history of this article can be downloaded here:

- PR File 1. Peer Review History. DOI: https://doi.org/10.5334/jopd.89.pr1

**TO CITE THIS ARTICLE:**
Welzel, C., Boehnke, K., Delhey, J., Deutsch, F., Eichhorn, J., Kühnen, U., Dragolov, G., Hess, S., & Larsen, M. (2024). The Values in Crisis Project: A Three-Wave Panel Study in Germany and the United Kingdom. *Journal of Open Psychology Data,* 12: 1, pp. 1–11. DOI: https://doi.org/10.5334/jopd.89

**Submitted:** 02 February 2023    **Accepted:** 11 March 2024    **Published:** 02 April 2024

