## [Peer Review History. · Journal of Open Psychology Data]

Authors' response to the Reviewers

Reviewer 1

Reviewer's comment	Authors' response
In Figure 1 / page 3, the protective direction is defined in the main text, but not emancipative. For the naïve reader, it would be worth explicitly defining this component too. The sample applies to the distinction between the thriving and struggling groups as these do not seem to be explicitly defined using the same terminology in the main text.	We rewrote the paragraph such that it now explains the distinction between emancipative and protective values as well as the thriving and struggling population segments.
This comment is more stylistic, so I will leave the final decision with the authors, but the introduction reads to end suddenly. I think it would be more effective as a segue to move the "The major disruption caused..." paragraph from page 2 to the end of the introduction to pose the research questions, then the method would flow more naturally after these.	Thanks for the suggestion. Instead of restructuring the introduction, we added two sentences at its end in order to make a smooth transition to the next section.
In Study design / page 4, the authors cite the questionnaires the questions were adapted from, but mention some questions were taken from validated scales, while others were created for the purposes of the study. Was there a validation procedure for these novel items and were any scales/sub-scales adapted from their validated origins? Flake and Fried (2020) highlight how it can potentially be problematic to adapt scales without outlining the new validation procedure.	We are aware of the importance of working with validated instruments. The Principal Investigators of the project have backgrounds in comparative sociology, cross-cultural psychology, political psychology, social psychology, and quantitative social science methodology. They did develop items for large-scale comparative survey projects such as the World Values Survey, the European Values Study, the European Social Survey, and the European Quality of Life Survey. In preparing the scales for the Values in Crisis project, given the sudden onset of the Pandemic and the speed at which related events (e.g., government restrictions) were unfolding, we had to strike the balance between the required expedience for going to the field and the established practice in the field of psychometrics. We did not have the time needed to validate the novel items. We constructed the latter to the best of our knowledge and experience. In order to make this issue transparent to potential users of the data, we explicitly address it in the manuscript. Please note that the paragraph in question (originally on p. 4) has been moved to Section 2.5 Materials/Survey instruments.
In Sampling / page 5 , how much was the	The opinion research company considers this

Reviewer's comment	Authors' response
average incentive to participants, if available?	information confidential. With the permission of the management, they disclosed it to us: 2 Euro or 200 loyalty points. If the reviewer does not mind, we would prefer not to disclose the exact amount of the incentive in the manuscript.
In Sampling / page 6, what was the purpose of the refresher sample given the aim of the project was to track changes across time? I personally have not conducted a panel study, so this might be relatively common, but it would be worth explaining the rationale.	We had expected a smaller sample size in Wave 3 than the one achieved and, therefore, wanted to boost the sample size so that segmentations would be possible for cross-sectional analyses. We now explain this in the manuscript.
In Materials / page 10, it would benefit from a brief explanation of why some questionnaire elements were included in 1, 2, or 3 waves. It makes sense why some measures like personality would be measured once in wave 1, but less clear to the naïve reader why other measures would be included in waves 2 and 3, or just 3.	We now explain the rationale behind the changes in the questionnaire topics (see Section 2.5 Materials/Survey instruments).
In Quality control / page 11, naïve readers would benefit from a brief overview of the quality control procedures. Its fine to cite further resources for longer details, but it would benefit the reader to have a rough understanding from the manuscript alone.	Upon consulting with the opinion research company, we included the information received on this matter.
One thing I think could be clearer is which questions in the materials relate to each of the questionnaires in table 4. For example, Schwartz values are clearly labelled, but I cannot clearly see which questions relate to depression and anxiety in well-being. So, there should be a consistent and clear link between the data and questionnaire items, with the contents outlined in table 4.	The table now cross-references the topics and the questionnaire items for each wave.
The data are not currently available in a repository; therefore, I will not be able to recommend the manuscript be published until there is concrete information on where readers can access the data.	The data and the accompanying documentation have been archived with GESIS and are already available to potential users: https://doi.org/10.4232/1.14148
The data are currently shared as a Stata file. While some statistics software such as R and SPSS can import Stata data files, the data are not currently available in an open, non-proprietary format. If the data files can also be saved as .csv files in addition to the Stata files, it would meet this criterion and be usable	The data have been archived with GESIS in csv-format, too.

Reviewer's comment	Authors' response
alongside the codebooks.	
My one recommendation would be to include a README file when the data are uploaded to a repository as you need to cross reference the file names with the manuscript, so it would be useful to have a document in the repository explaining what each file contains.	The data files and documentation on the website of GESIS is well structured. We believe users should have no difficulties to orient themselves.
The authors explain in the manuscript the data have been anonymised and they signed informed consent forms. However, I cannot see any mention of where the study was granted ethical approval. Therefore, the authors should explain – probably in section 2.7 on page 11 – where the study received ethical approval and include the reference number if available.	As per the guidelines of the German Research Foundation (DFG) on research projects in the Humanities and Social Sciences, a statement by an ethics committee was not required. We now mention this explicitly in the manuscript.

Reviewer 2

Reviewer's comments	Authors' response
The data must be deposited under an open license that permits unrestricted access (e.g. CCO, CC-BY). Data access is currently limited as outlined in sections 3.6 and 3.7 of the manuscript.	GESIS supports open access. We have, however, restricted the use to academic research and teaching. In light of the remark that the license should permit UNRESTRICTED access, we ask for your understanding that we will not open the data for any use, especially commercial.
Data is currently available in Stata format (dta) only. The authors might simply consider to provide an additional version of their data in a non-proprietary format (csv etc.) to address this issue.	The data have been archived with GESIS in csv-format, too.
Section "(3) Dataset description and access" needs to be updated as data seem to be available now.	Done.
Section "(4) Reuse potential": I would like to encourage the authors to further elaborate on their ideas for data reuse in this section. Certainly, they will have some more specific exemplary ideas on data reuse scenarios that might inspire future users of their data. Additionally, they might consider to outline the strengths and/or limitations of their data in this regard.	We would prefer not to offer specific ideas for data reuse. We would like to be the first who would pursue those ideas that we have in mind. As to the strengths and limitations, we address some in other sections of the manuscript.